# The Sacred Building and the City: Decoding the Formal Interface between Public Space and Community

João Silva Leite *, Sérgio Fernandes and Carlos Dias Coelho

CIAUD, Research Centre for Architecture, Urbanism and Design, Lisbon School of Architecture, Universidade de Lisboa, Rua Sá Nogueira, Polo Universitário do Alto da Ajuda, 1349-063 Lisboa, Portugal; sergiopadrao@edu.ulisboa.pt (S.F.); cdcoelho.luotp@gmail.com (C.D.C.)
* Correspondence: joao.leite@edu.ulisboa.pt

**Abstract:** The reflection on sacred places continues to assume significant relevance today in urban space production. The public value of sacred buildings has consolidated over time an aggregating sense of community, representing spaces for meeting and sharing. Their historical relevance as spaces for meditation represents for mankind places of personal reflection, while they have always played an important role in the city and in its symbolic and spatial structure. Thus, starting from the hypothesis that the sacred space is affirmed as an interface, because it welcomes the individual and serves the community, we examine the architectural features that enhance this ambivalence, exposing transition systems between private and collective spaces, seeking to systematize essential composition matrices for new urban spaces for public use. Assuming Lisbon as a framework, this article proposes a comparative reading between two paradigmatic buildings—Sagrado Coração de Jesus Church and the New Mosque of Lisbon—with similar goals according to the relationship between architecture, place sacrality, and the urban public space. Methodologically, drawing is used as an interpretative tool and, through formal decomposition, this article tries to demonstrate that these buildings are the result of a reflection deeply determined by the value of the place's identity in the city's public space system. According to these case studies, sacred buildings are conceived based on formal and spatial links that are rooted in Lisbon's urban layout. It is sacred buildings that are at the origin of urban places for public use. Each one of these buildings share an idea of architecture with an urban and public role which integrates the objects with the shape of the city and contradicts the tendency for the dissociation between urban elements. In a way, they can be considered paradigmatic examples of architecture with an urban vocation.

**Keywords:** sacred building; urban architecture; public space; interface space; passages; Lisbon





## 1. Introduction: Urban Architecture and Sacred Buildings

*"The unique relationship between the open area of the square, the surrounding buildings, and the sky above creates a genuine emotional experience comparable to the impact of any other work of art".*

(Zucker 1959, p. 1).

Sacred spaces have always played a relevant role in humanity in the construction of a meaning between Man, his spiritual intimate and a specific territory. Through architecture, the sacred space acquires a body and physically and emotionally links mankind to a place. A tension is created between the value of the building as an object and the relevant meaning it establishes within the urban space. The architectural object, through its formal features, constitutes spatial atmospheres that create conditions for individual permanence and reflection, but also for communion with the collective. The sacred building assumes a public significance, a place of meeting and to be with others.

This collective sense[1] has, on the other hand, echoes in the public space around the sacred building. The public space is constituted as a moment of transition that connects the

building and the city, becoming an important piece in the ritual of articulation between the community and the sacred place. The public space builds upon the visual approach and spatial referencing systems (within the urban space), intensifying the symbolic value of the place and imprinting an urban function on the sacred space. In the study of Italian cities carried out by Camilo Sitte in 1889, this dichotomy between religious/sacred spaces (church) and squares is particularly evident, showing, on the one hand, the morphological diversity of the phenomenon and, on the other hand, exemplifying how the public space values the church as an object, while at the same time suggesting that its very position allows it to take up significant spatial wealth in the city.

Furthermore, public space as a more abstract idea finds continuity in the interior space of the architectural object. The collective/communal sense of the religious space dilutes strict limits, constituting in different situations moments of extension and blending between the interior (private) and exterior (public) space. This ambiguity was, to a certain extent, already exposed in 1748, in Nolli's plan of Rome, demonstrating the deep connection between public spaces, namely squares, and sacred spaces, namely churches. In this case, churches appear as natural interfaces for building a community space that serves the collective but also the individual and their spirituality.

This article proposes a reflection on how sacred architecture can be understood as a didactic topic in the construction of public buildings that open up to the city, interconnecting in its structure with public space. To this end, it uses as a support part of the material produced by the research project *Building Typology: Morphological Inventory of Portuguese City*[2] developed within the Lisbon School of Architecture, Universidade de Lisboa, as a methodological and conceptual source. Thus, the examples of the Church of Sagrado Coração de Jesus, a building designed by Nuno Teotónio Pereira and Nuno Portas[3] in 1970, and the New Mosque of Lisbon, a project conceived by Inês Lobo[4] in 2013, both located in Lisbon (Portugal), are taken as paradigmatic situations of sacred spaces that start from the idea of an empty space with a public nature, which structures the entire building's architectural composition.

These two case studies aim to demonstrate the building's potential as a producer of public space, particularly when it has a public value and serves the city. The intention is to show the architectural qualities of the two buildings and how, regardless of the religion they serve, they build public spaces. The Sagrado Coração de Jesus Church and the New Mosque of Lisbon are interpreted as didactic objects for future architectural productions. Architectural production is understood as a support for the collective life of the city.

This article begins by synthesising the methodology and material base that supports it and then reflects on the historical association between religious buildings and urban space. This dichotomy has questioned the spatial limits of public space in several examples, and it is therefore important to discuss how it has redefined the space for collective use. To this end, the two aforementioned case studies are taken as concrete opportunities to observe this phenomenon, regardless of the religion behind the building or the time of its construction. This structure aims to contribute to the debate on the value of religious buildings in the production of public space, but also, through this typology, to reflect on the potential that buildings with a singular character can hold in the design of urban form and in the way we can inhabit cities.

## 2. Materials and Methods

This article was based on two intertwined research projects. On the one hand, it was based on the typological inventory produced by the *formaurbis* LAB research group, a laboratory included in the Research Centre for Architecture, Urbanism and Design (CIAUD) at the Lisbon School of Architecture, Universidade de Lisboa and, on the other hand, on an individual post-doctoral research project with the theme "in-between in contemporary architecture"[5].

Through the Building Typology research project, the *formaurbis* LAB research group completed a morphological inventory of the Portuguese city. Through exercises of read-

ing and decoding the Portuguese urban fabric, more than 120 cities were inventoried and the various constituent elements of the urban fabric–square, street, block, plot and building—were studied. The Building Typology project dedicated itself to studying and systematizing the different Portuguese typologies, sorting them based on three theoretical categories—Programme, Context, Time[6]—organized by territory mapping and the selection of 120 buildings. Each building (framed in the three categories mentioned) was a case study, representing a typology. A set of comparative tables were added to each of these buildings in order to present the formal diversity of the typology. In this way, the inventory[7] of 120 studied buildings was expanded to more than 700 case studies analyzed comparatively (Figure 1). With this universe of case studies, this article focused on the Context category, which is particularly aimed at studying typologies whose principles are profoundly determined by the circumstances of a context—natural or built (city). In this category, we can observe a set of built typologies that have the particularity of designing the city. Streets, squares or passages are some of the situations created by buildings and which shape or mold the public space of our cities. Based on this inventory, two cases were analyzed, namely a church and a mosque, two religious buildings that affect the design of public spaces, including passage and square. Both are born from the design of public space (Figure 2).

**Building Typology**

*[ three categories ]*

Programme

Context ........................ Land .......... ridge
                                              slope
Time                                          rock
                              Water.......... on water
                                              waterfront
                              Greenery
                              **City** .......... **square** ...................... *New Mosque of Lisbon* *
                                              street
                                              intersection
                                              **passage** ...................... *Church of Sagrado Coração de Jesus*
                                              urban block
                                              plot
                                              infrastructure

* *The case of the New Mosque of Lisbon was not included in the final list of cases studied in the inventory produced by the Building Typology research.*

*The fact that the building had not yet been built was a condition for its non-inclusion, as the field of study focused on built buildings. However, the project for the New Mosque of Lisbon is an enrichment for the debate on this type of building, and is therefore part of a very extensive universe of other referenced cases*

**Figure 1.** Building Typology, three categories: case studies' identification within the research classification structure.

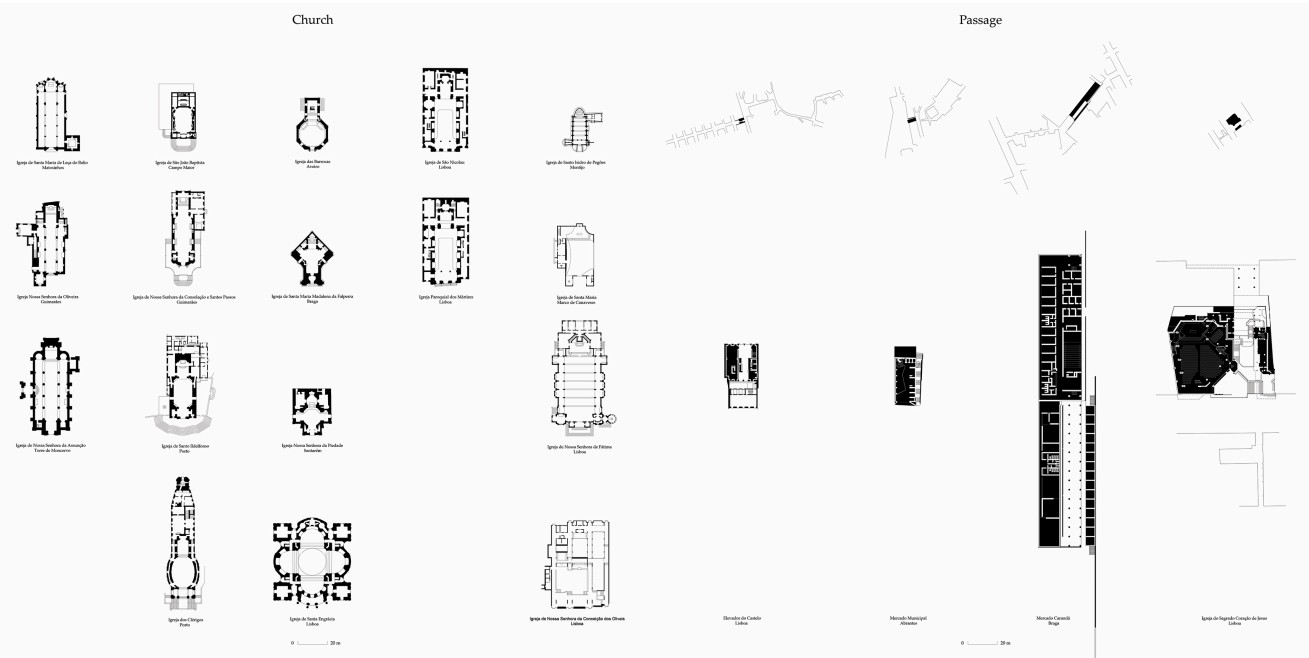

**Figure 2.** Church comparative table and Passage Buildings comparative table.

It is also important to note that in conjunction with this "Building Typology" research, the post-doctoral research organized a list of examples of architectural works built in Portugal that show a clear concern for designing public or semi-public transitional spaces that seek to build integration spatial systems between building and the urban layout. This list was partially supplied into the "Building Typology" inventory and includes the case studies discussed in this article—Sagrado Coração de Jesus Church and the New Mosque of Lisbon.

Methodologically, this article was based on morphological and spatial decomposition (Christ and Gantenbein 2012; Dias Coelho 2013; Eisenman 2008; Kaijima et al. 2001; Panerai 1999), systematizing through drawings a set of elementary characteristics and systems of spaces that can demonstrate architectural qualities and also that can be transferred to other architectural programs with a public and collective nature. Through the analytical exercise of decomposition (Borie et al. [1978] 2006; Correa 2018; Gandelsonas 1999; Geist 1985; Lampugnani et al. 2019, it is possible to isolate architectural or urban elements (Foscari (2014); Koolhaas (2018); Krier (1983)), making it clear which tools were used to build the supports for urban life (Gehl [1971] 2017). On the other hand, the process of decoding the two examples also sought to highlight the system of spaces configured by the two buildings. This made it possible to visualize the public space generated, which parts and elements constitute it and how it relates to the different functions or services offered by the building.

The case studies were assumed as pedagogical references in the way they constitute project solutions that promote more communal and shared urban living. This study sought to contribute to building an architectural design lexicon capable of supporting contemporary production and disciplinary teaching, always with the aim of highlighting the unique role of the public building as an interlocutor between an intimate (private) and collective (public) space.

## 3. The Sacred (Building) and the Public (Space): Host, Permanence and Transition

Architectural objects with a sacred meaning, due to their spiritual and religious relevance, appear in the urban fabric in a way almost associated with a position of prominence or notoriety (Fernandes 2014). The symbolic and referential sense is exalted in urban space not only by the objectual singularity of the building or typology, but also by the dialogue it frequently establishes with the public space.

Although this relationship has an almost ancestral meaning, whether in an urban context or in a territorial and isolated context, in the cases of Greek religious complexes, such as a temple of Athena in Pergamon, built in 2.° century B.C., these spaces served as the starting point for the layout of the precinct, where the relationship between the sacred building and the public space was the result of a new urban framework or the stoas that were built after the temple (Doxiadis 1972). Also, in Roman temples and their connection to the forum, or even the small caves or hermitage chapels where the spirituality is strongly linked to nature[8], it is verified that it is with the emergence of Renaissance thought that this relationship between building and public space is recovered. In Western Christian cities, the dichotomous nature of public–private spaces is clear, unlike other cultures or times, where sacred spaces asserted themselves as their own restricted nature, as seen in examples from Ancient Greece and Japanese Shinto shrines[9]. In this sense, a close link between churches and public spaces was consolidated in the Christian European city.

"*The main squares of each city were indispensable for its daily existence*".

(Sitte [1889] 1980, p. 160)

The process of transforming the churchyard into a square for Notre Dame Cathedral in Paris (Figure 3), Milan Cathedral, and even the Portuguese case of Porto Cathedral provides paradigmatic examples that "allow us to visualize the action of a building of great collective importance on the surrounding public space" (Dias Coelho 2002, p. 175). This process that takes place over several centuries exemplifies a clear intention to value the religious building at the same time that it generates conditions in the public structure of the city to create a space capable of hosting and articulating the community and the building with a sacred meaning. The spatial system that is produced, comprising the square or street which connects us or leads us to the religious building, constitutes a progressive transition between the public and sacred space, a place with greater intimacy.

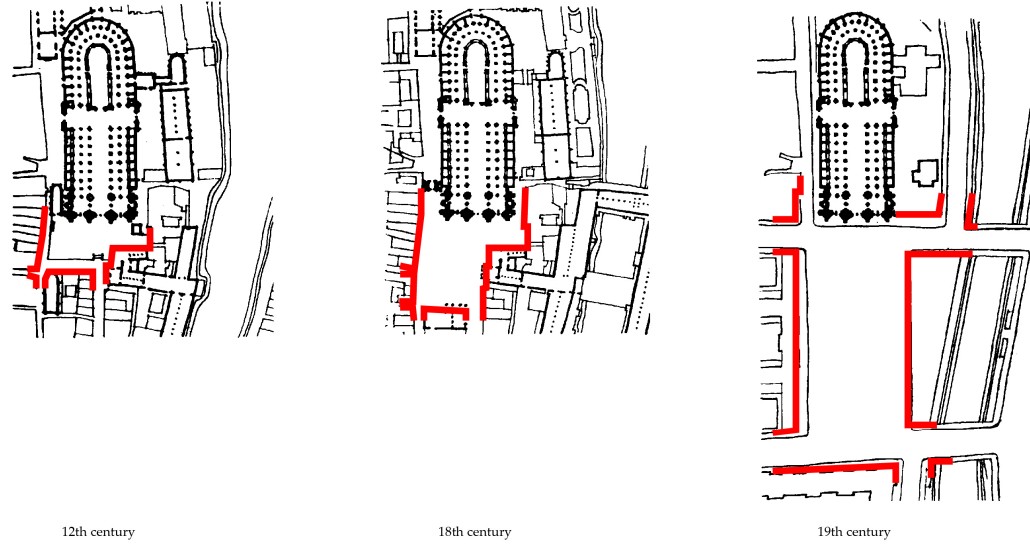

12th century          18th century          19th century

**Figure 3.** Transformation process of Notre Dame square, Paris.

The square appears, then, as an interface device between people and the building. It assumes itself as a place that marks the church, displays a symbolic object, but also a meeting place, a locale that people assign *permanence* and that precedes the passage to the interior space of meditation. The public space is, therefore, an important instrument of *transition*, constituting one more piece in the system comprising the sacred place and the community.

The urban operations of the 16th century, carried out by Pope Sixtus V, reveal in an extreme way this dichotomy between urban spaces and religious buildings. In addition to the squares that dialogue with the imposing churches, a new stratum is imposed on the

city that takes the religious building as a landmark that guides the pilgrim and orders the entire city. On the other hand, with the rise of the Baroque as the dominant architectural language, it appears that the formal dialogue between church and square, or axial system, is developed to a new dimension. The formal composition of the main facade of the building is exacerbated by the formal characteristics of the public space that surrounds it during this period. The use of perspective as a design tool allows for establishing visual connections or spatial sequences that intensify the interconnection between object and space. It is also during this period that the facade acquires a certain thickness, consisting of several architectural elements, such as columns, bases, unevenness, porticos or galleries, which contribute in a decisive way to enriching the system of transition and articulation between the inside and the outside. It constitutes a threshold space that welcomes the collective and prepares the passage of the individual to the sacred place.

## 4. The Threshold as an Urban Device: The Public–Private Interface

In the close relationship between sacred space and public space, the threshold space becomes evident as a particularly relevant device in the transition between exterior and interior. This spatial fringe is assumed as an important moment, on the one hand, in the recognition of approaching and entering the sacred space and, on the other hand, in the construction of small places of permanence. Courtyards, patios, small variations in the level of the pavement, stairs, benches, cover structures, or other architectural elements contribute to the construction of a thick threshold that articulates the exterior–interior interface, preparing the user of the urban space for the transition into a space of spiritual reflection, at the same time as offering conditions for permanence. The limit between the interior of a religious building and a public space is not defined as a simple barrier to be crossed, but rather it becomes a place, a space with a form that offers conditions for appropriation and that becomes habitable (Van Eyck [1962] 2008). The threshold space emerges as an interface between public and private, merging spaces and thus contributing to the construction of an idea of collective space (Boettger 2014). The threshold space assumes an ambiguity of transitions, making the limits where the interior and exterior space begin or end perceptible and, at the same time, affirming whether it is private or public property[10].

To a certain extent, this spatial value was also identified by Alison and Peter Smithson (1974) in their text "The Space Between", published in *Oppositions* magazine, where the in-between space is understood as an opportunity to rethink new forms of spatial relationship between the built and the city (Juárez Chicote and Ramírez 2014).

Although the text is more than 40 years old, the reflection of these British architects is extremely current. The idea of exploring a thick and habitable threshold in the design of the city asserts itself today as a device of spatial and architectural composition that contributes to the integration between urban elements, building and square/street (Silva Leite and Proença 2020), dissipating rigid boundaries that block greater fluidity between the various types of space. The symbiosis between public space and sacred building is consolidated and a system of integrated and inseparable spaces is constituted.

> "*Physical urban quality is in the measure, the proper understanding of the limits of a space. As soon as we define it, we segregate it. Good public space has no limits*".

> (Solà-Morales 2010, p. 31)

## 5. The Sacred Building as a Generator of Public Space in Lisbon: Sagrado Coração de Jesus Church and the New Mosque of Lisbon

In Lisbon's urban form, the permanent dialogue between public space and religious places, consisting mostly of Catholic temples, finds echoes through various spatial relationships (Figure 4).

# Church . Public Space

Time . Space

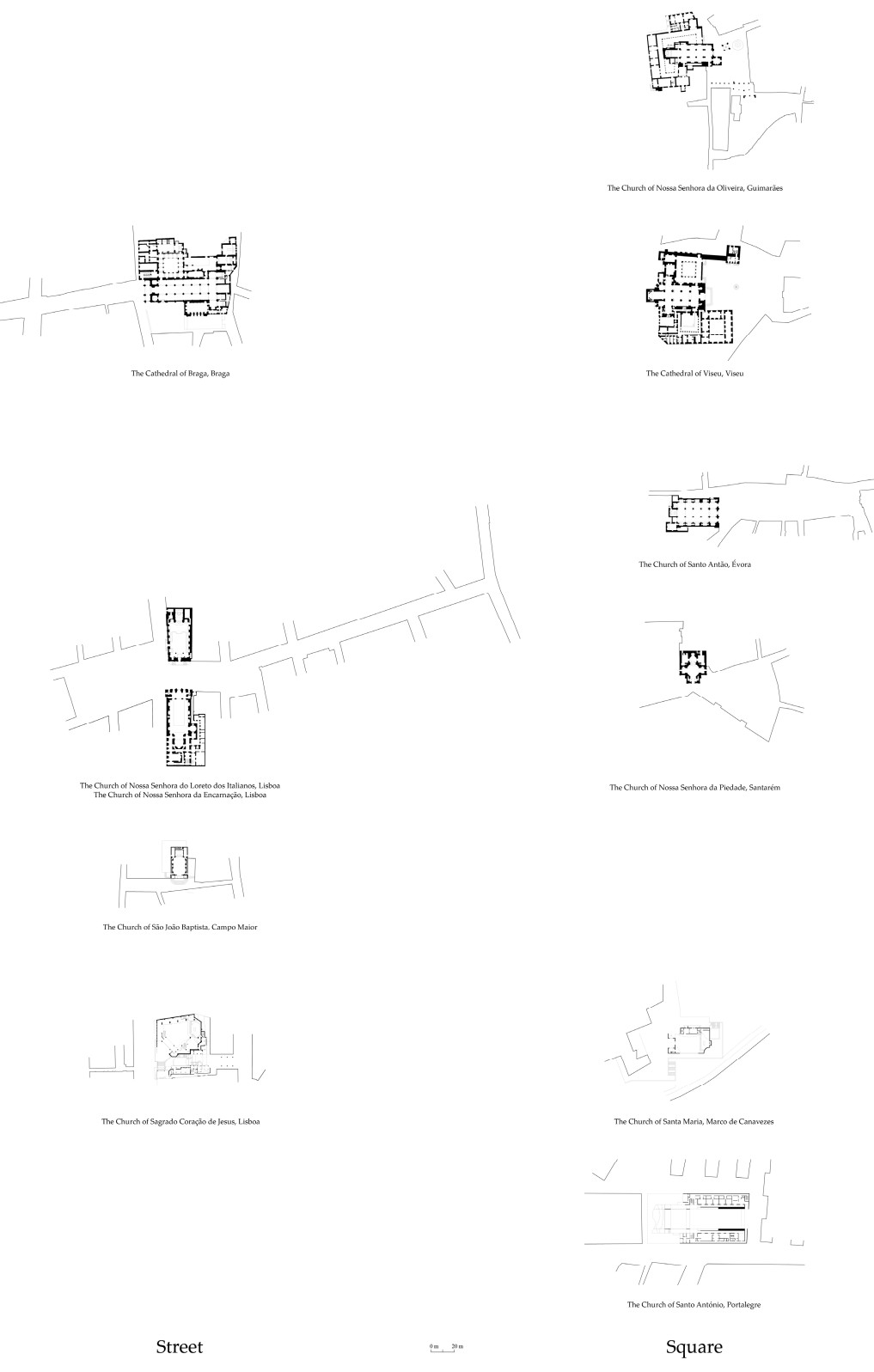

**Figure 4.** Comparative table: Some relationships between churches and public spaces (squares/streets) in the Portuguese urban context. Through an axial organization, the examples are ordered by time (oldest at the top) and space, street (**on the left**) and square (**on the right**).

The church–square relationship is, as a rule, the most observed, expressing more clearly the symbolic value of the architectural object before the public space and the city.

However, there are examples where the compositional relationship is established in a more axial way, having the street as an anchoring element, and where the positioning of the church seeks a more anonymous integration into the built fabric. Even so, one can observe the existence of transitional spaces, such as small churchyards or staircases that mark the moment of approach to the object and establish a space for the community to meet in the urban fabric.

Following the Second Vatican Council (1962–1965), there are some examples of religious architectural production in Lisbon that seek to open up the religious space to the community. Cases such as the Sagrado Coração de Jesus Church (1970) and the Parish Church of Nossa Senhora da Conceição dos Olivais (1988) are examples of authentic religious complexes where various activities are brought together to expand the sacred space in order to serve the community in a deep collective spirit.

In this respect, the Sagrado Coração de Jesus Church, built between 1962 and 67 but only inaugurated in 1970, represents a milestone in the production of religious spaces in Portugal, not because of its formal and organizational characteristics of the space but mainly because of the way it connects to the city and to the urban layout.

From this work, several examples followed, understanding the empty space around the religious complex as a place for the expansion of religious activity itself and as a space for connection to the community in a broader way. In this sense, it is important to look at this case, understanding the formal links it establishes with the public space, namely in the way it affects the design of a street, and to simultaneously observe the recent project of the New Mosque of Lisbon, regarding the way in which it also defines a collective place based on an interior square. Although this building has a different religious foundation, it has the same public vocation and has developed a strong relationship with the public structure of the city.

### 5.1. Sagrado Coração de Jesus Church

Sagrado Coração de Jesus Church, a project by the architects Nuno Teotónio Pereira and Nuno Portas, both progressive Catholics and belonging to the MRAR[11], shows a deep sense of progress influenced by two important events—the aforementioned Second Vatican Council (which redefined a new relationship between the Eucharist and the community) and a certain democratic spirit that was beginning to effervesce in Portuguese society. The interior space of the church assumes the configuration of an amphitheater, a large audience, and the religious building constitutes a built complex that constitutes an idea of "microcity" (Grande 2021).

It is precisely this urban sense of the church that makes it a truly unique case in the Lisbon context. Inserted in the middle of the urban block structure, this church seeks a certain discretion, integrating itself into the urban fabric and privileging relations of continuity with the urban layout of the city. This sense is also reinforced in the conceptual process of the formal composition of the entire building complex. The building is born from a massive block (Proviência and Baía 2019) from which a passage is theoretically subtracted, which assumes the double function of, on the one hand, fragmenting the block into several volumes and, on the other hand, establishing a transversal connection in the plot[12], connecting two streets that delimit the urban block where the church is located.

In this way, this passage directly determines the architectural shape of the entire religious ensemble, while at the same time giving it a centrality and a symbolic meaning (Figure 5).

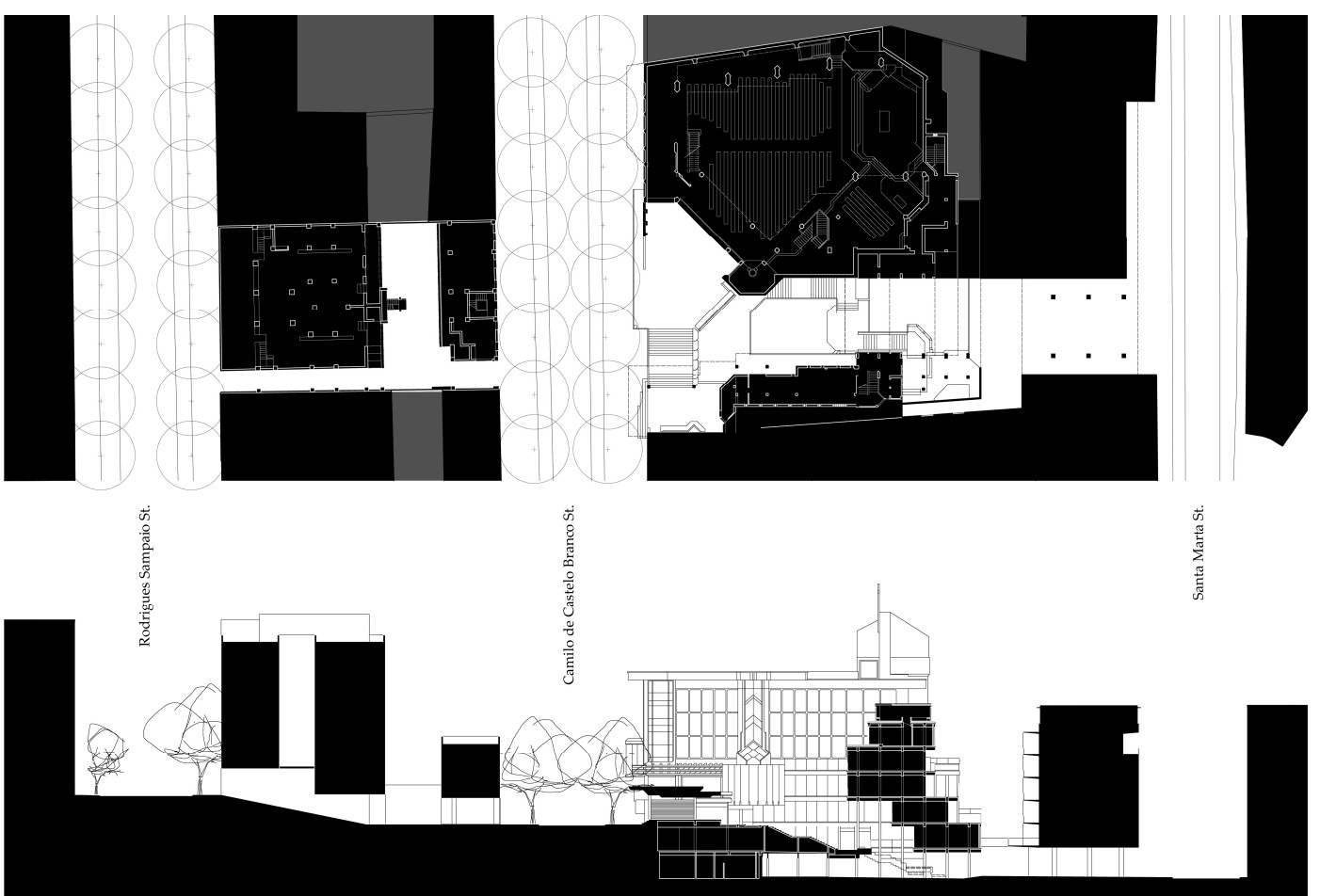

**Figure 5.** The passage produced by Sagrado Coração de Jesus Church: plan and section.

The importance of the passage is such that the various activities and functions that make up the entire complex are added to it. The passage embodies a public sense of the street, a gravitational line where community and collective spirit meet.

This empty space is the structuring and supporting element of the building's religious life, but also an element that ties the project to the city. This passage. which articulates a unevenness of about 7 m between the two streets, assumes itself as a street–staircase that dilutes limits and integrates into the urban layout and the daily dynamics of the city and the people of the neighborhood.

However, the relevance of the passage in the relationship between church and city is not synthesized only in the connection that it provides, or in its relevance as a void in the volume composition. The morphological features themselves and the various architectural elements that compose it configure a system of spaces and sub-spaces that are discovered along its path (Santana and Cunha 2020). The shifts between darker, open and wider spaces constitute spatial composition strategies that give human scale, factors of surprise or discovery that lead the user to different uses or religious spaces (main hall, mortuary chapel or study rooms and community meeting), or simply invite people to remain in the open space, sitting on the handrails between the stair platforms, on a bench or on an esplanade that extends from a restaurant area (Figure 6).

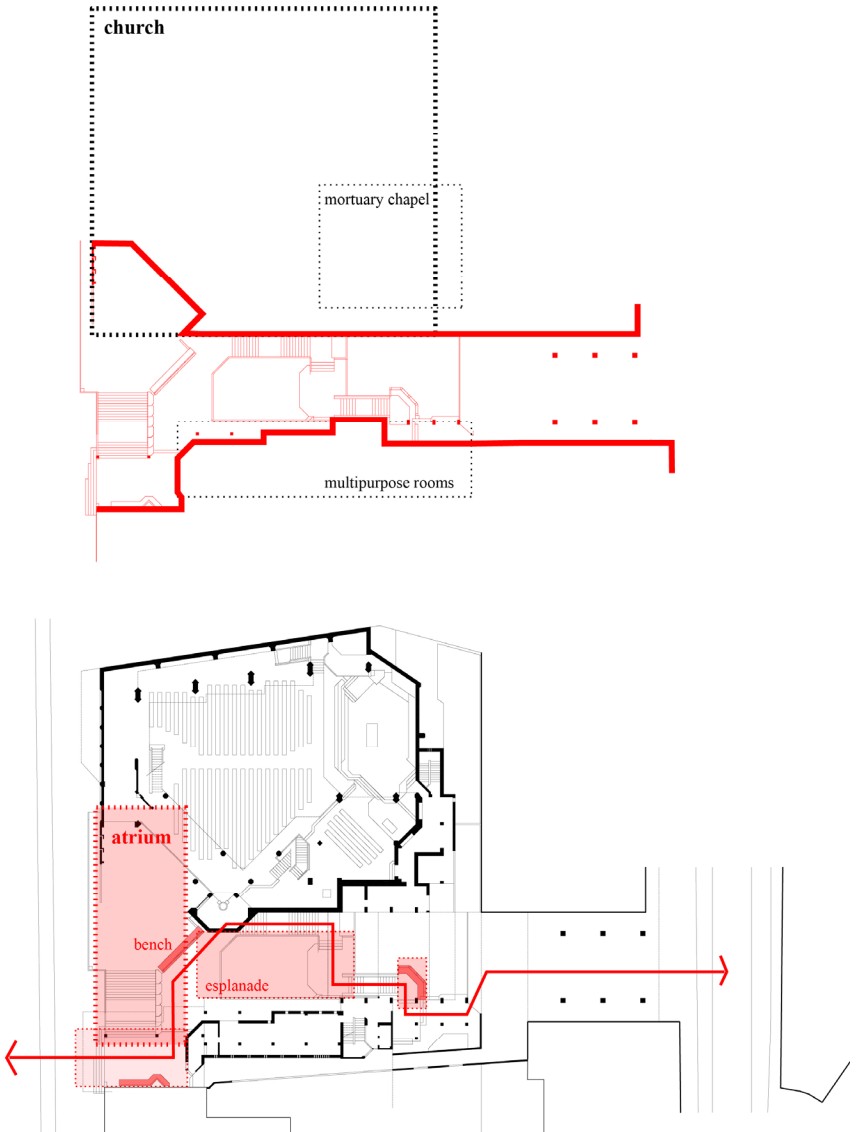

**Figure 6.** Sagrado Coração de Jesus Church passage system: collective places, permanence elements, functions.

This constant dialogue between the passage and the building is also particularly evident in the way that the architects designed the entry point. It opens into a small churchyard, connected through the passage, but not directly to the city's public structure. It is through the passage that it is possible to enter the sacred place. However, it is a twist in the main façade that announces the passage and indicates the access.

This detail expands the public and structuring sense of the passage. Another determining aspect of the passage is its domestic spirit. Despite its clear public vocation, the passage preserves intimacy, a certain idea of privacy, where the community finds a place to meet and share. This atmosphere is reinforced by the volumetric dialogue that is composed between the void and the built, in a sharp contrast created by the proximity between the different bodies of the architectural object. Complementarily, the certain verticality of the space will collaborate for a reading of a more contained, restricted, private place. It is in this ambiguity that the collective sense flourishes, fulfilling the desire expressed by the architect Nuno Portas to the Arquitecturas magazine in 1971, where he says that he would like the space to "stimulate activities where the sacred becomes profane and the profane becomes new sacred"

The passage welcomes the city's public activity, making it a natural extension of the religious space, making the intimate and the collective compatible in a complex and stimulating spatial symbiosis (Figure 7).

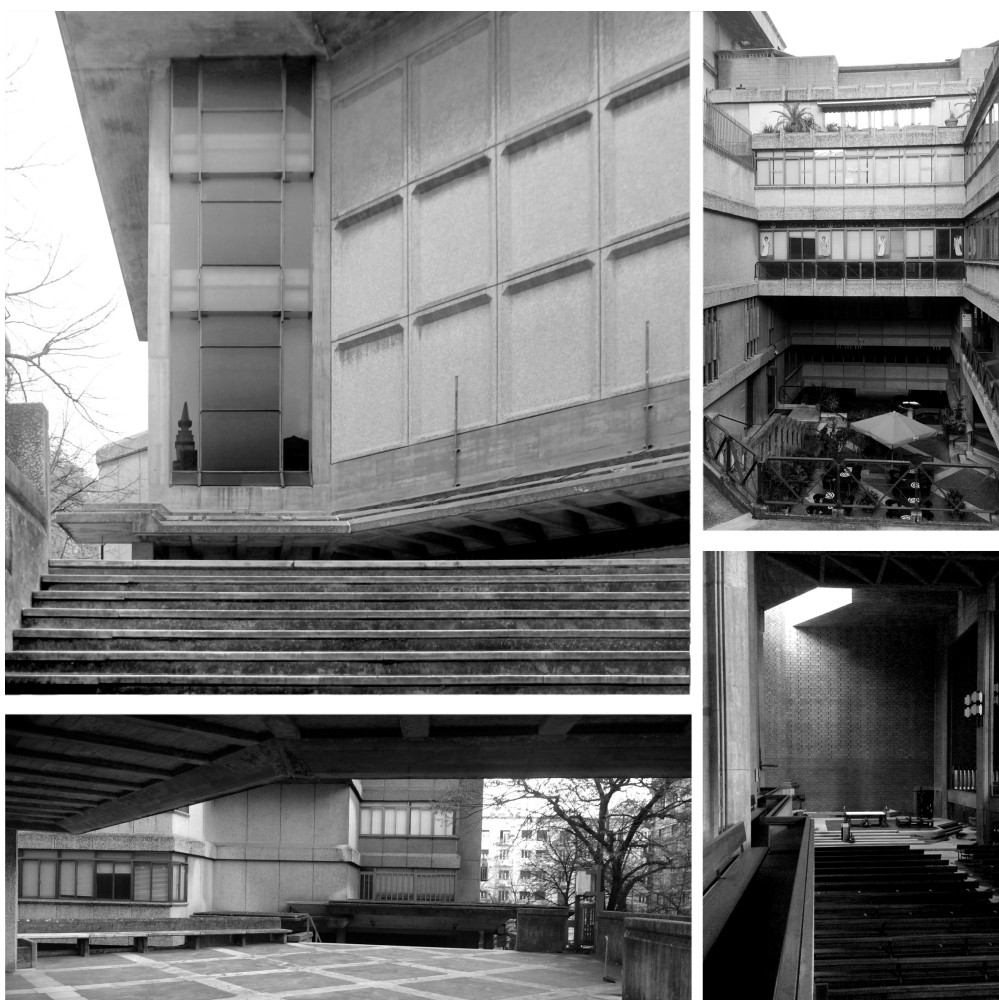

**Figure 7.** Sagrado Coração de Jesus Church photographs. Source: João Silva Leite, 2022.

### 5.2. New Lisbon Mosque

More recently, a project a new mosque appeared in Lisbon, which ignited a public debate on the relevance of a religious building as an aggregating element of a community, thus allowing its progressive integration into Lisbon's civitas. The project was promoted by the Municipality of Lisbon in collaboration with the local Muslim community from Bangladesh in order to respond to the growing needs of the Muslim community in the eastern part of Lisbon, particularly in the Mouraria neighbourhood[13]. The proposal developed by the architect Inês Lobo (2013) started from a clear and predetermined idea of building a passage between two important streets in the Mouraria area of the city—Palma Street and Benformoso Street.

In this way, the proposal responded to an old desire of the residents of the neighborhood who, since the 19th century, have called for a new connection between Mouraria and Lisbon downtown.

With the expansion to the north of Palma Street through the opening of Almirante Reis Avenue, the segregation between the neighborhood and the west side of the city was accentuated, generating numerous social problems.

It is within this framework that the project sought to act. Starting from a desire to draw a large void between the dense built fabric, the New Mosque of Lisbon is structured

around a square that, through two levels, establishes the articulation between the two streets (Figure 8). The new square, delimited on its sides by the built structure, but opening to the streets, asserts itself as a space of continuity of the urban layout, constituting itself as an element of connection and, simultaneously, as a place of aggregation and permanence[14].

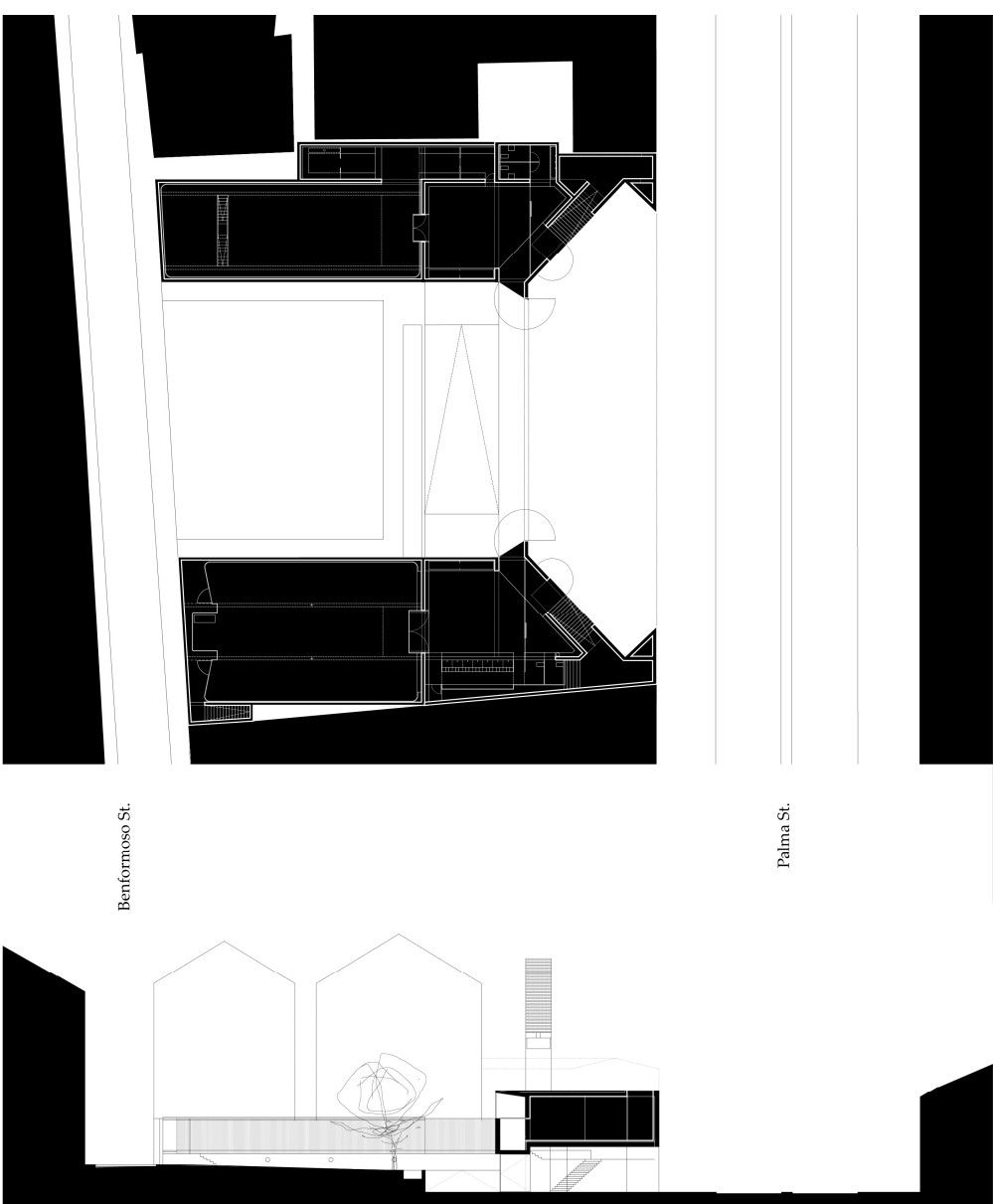

**Figure 8.** The New Mosque of Lisbon passage: plan and section.

The public square is assumed in this project as a reinvention of the traditional courtyard of Islamic mosques, which have a semi-public nature and which normally play an important role in the transition between the (public) city and the sacred space for prayer. This typological transformation from patio to square intends to open up this space to a wider community, helping to integrate this often stigmatized and segregated population. Thus, in an attempt to centralize the spatial relevance of this public space, the built complex is then organized around the void, having a varied programmatic base that does not just pass through the sacred place. The building contains, in addition to prayer and meditation spaces (male and female), a set of community support rooms, such as a refectory and a communal kitchen, and even two multipurpose rooms, with the intention that one can be used as an art gallery (Figure 9). These uses, which go beyond exclusively religious spaces,

play an important role for the community. The kitchen and refectory open up to a wider community, serving the entire population of the neighborhood, whether Muslim or not.

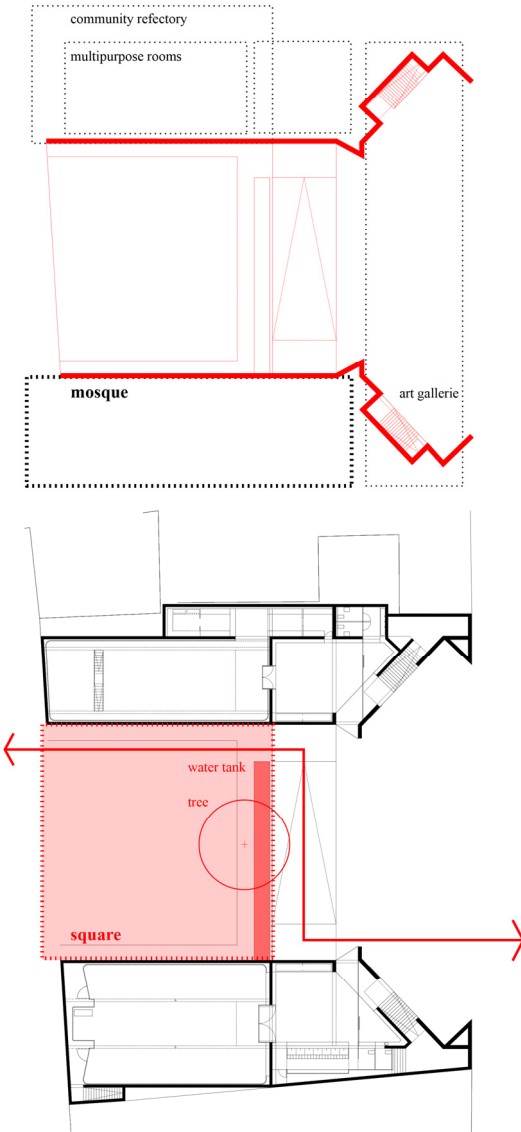

**Figure 9.** The New Mosque of Lisbon passage system: collective places, permanence elements, and functions.

This collective sense is an important step in the creation of infrastructures that collaborate towards better social integration of the different populations and a certain identification with the place. The public sense of the square proposed by the mosque's design is echoed in the mosque's interior space. The public sense of the building transcends the physical limits of the building itself.

As in the case of Sagrado Coração de Jesus Church, in the project of the New Mosque of Lisbon, it is also possible to identify a set of formal solutions, architectural elements and programmatic strategies that intend to create the public place, the space capable of asserting itself as a new point of gathering of the collective, comprehensive and multiracial. The architectural object, due to its formal and spatial configuration, becomes a true interface between public and private spaces.

Project options such as raising the building from the ground, in front of Palma Street and opening the square to Benformoso Street, enhance a natural continuity between the two streets, smoothing the transition of elevation through a system of ramps that makes

all pedestrian circulation comfortable and quite fluid. In addition to the existing volume (elevated) on Palma Street, it contributes to the construction of the portico (Figure 10), attributing human scale to the passage and constituting continuities in the pre-existing street front alignments. On the other hand, the water tank and the tree placed over the square, on the side of Benformoso Street, and the distribution of the main uses on the side bodies of the building enhance the recognition by people of the space as a place of permanence and social meeting.

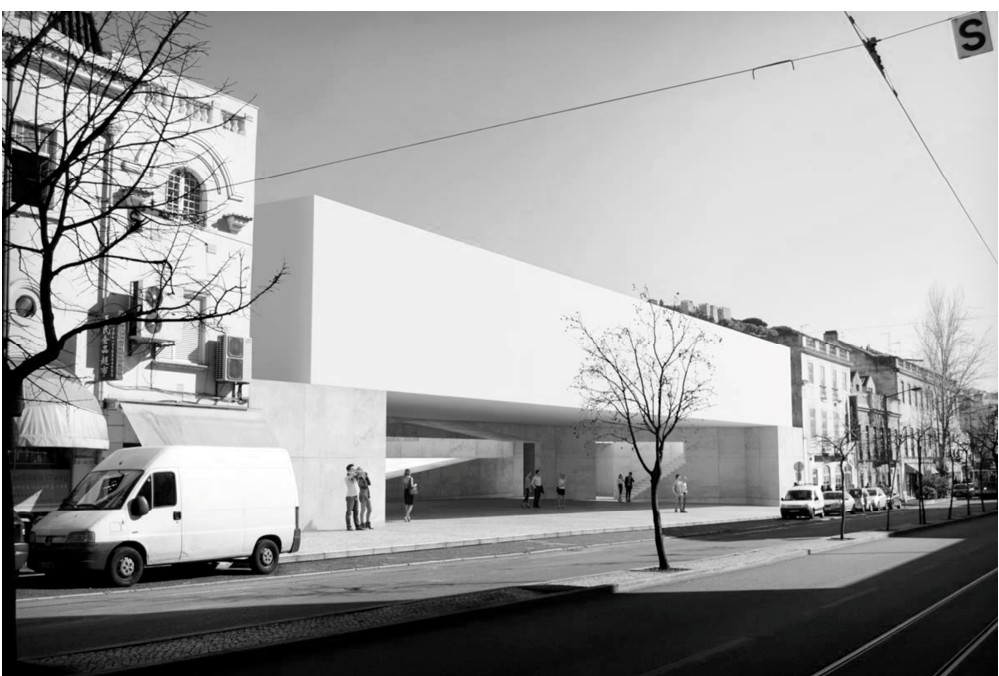

**Figure 10.** The New Mosque of Lisbon render. Source: Inês Lobo Arquitectos, lda. (2013).

## 6. Discussion: Still Learning from Sacred Building Typologies

> "*Of all the architectural elements, the wall comes first. The primary purpose of a wall is to establish a relationship. Association comes before separation*".

(Khosravi 2021, p. 67)

The sacred building has recurrently been used as a reference over time, and the history of architecture reveals that the compositional characteristics of these buildings have sometimes been reinterpreted and other times transferred to give rise to buildings with other architectural programs. On one side, we can place Hagia Sophia[15] and its main dome, which was successively reinterpreted and was the main reference for the mosques designed by de Mimar Sinan[16]. Meanwhile, on the other hand, we can consider some of Andrea Palladio's villas as a rehearsal of transfers of the characteristics of sacred buildings to the intimate sphere of housing.

Through a parallel look at the two case studies, Sagrado Coração de Jesus Church and the project for the New Mosque of Lisbon, it becomes possible to reinforce the relevance that religious/sacred buildings have in the spatial composition of the city and in the potential to generate places of social aggregation, community and collective identity, in addition to the symbolic and referential meaning that they have has always represented in the urban context. Although the two studied buildings are intended for different faiths, they both produce new public spaces in a very clear way. Both buildings shape the urban structure and are designed as objects based on this characteristic. In the case of the mosque, the creation of a public square constitutes a excellent collective meeting space, both for Muslims and for any other inhabitant of Lisbon. The square, as a reinterpretation of the private courtyard of a common mosque, becomes an interface for sharing and interaction between

everyone[17]. The Sagrado Coração de Jesus Church, on the other hand, creates a more complex system of public space. Along a passage, several places invite people to stay and socialize. Larger spaces, such as the churchyard, and smaller spaces such as a simple bench emerge as devices to encourage the collective sense. This public space structure is therefore more than just a crossing device—it is a place that seeks to serve as an extension of religious activities and also a meeting place for the entire neighborhood community (believers or non-believers).

The formal decomposition of the two cases made it possible to highlight the role that public buildings and specifically sacred buildings can play in contemporary city design and in the design of public spaces. Public program architecture therefore assumes an additional responsibility in the production of collective urban space, not referring only to the symbolic or scenic meaning of the building. Through decoding religious buildings' formal structure, it is possible to stabilize more general compositional principles that could be transferred to the production of other architectural pieces, especially those that have a public character. Thus, as religious buildings assume a structuring relevance in the city, not only due to their symbolic character, but also because they are closely associated with the production of public space, it is important to explore parallels and question the possibility that other typologies of public buildings can incorporate similar formal features, which allow the construction of an identical cohesion between public buildings and public space to that which occurs in the relationship between buildings and cities as established by sacred buildings.

It is perhaps important to recall the thoughts of Manuel Solà-Morales who emphasizes the strategic value of producing public space through the use of private elements that know how to interpret collective space as a factor of urbanity. Spatial ambiguity and the dilution of rigid limits make urban space more versatile and changeable to different circumstances, rhymes and times.

> "*Projects that collectivize. These projects, the most incisive, accept the strategic goal of creating public space with private ingredients, on the basis of an understanding of collective space (public+private) as a defining substance of what is urban. Explicitly or otherwise, such projects take the view that urbanizing means collectivizing, and they may have a lot or a little in terms of form, but they do not shape. Rather, they are actions of mental strategy*".

<div align="right">(Solà-Morales 2010, p. 29)</div>

## 7. Conclusions

The built typology with public vocation thus incorporates an additional responsibility. The building must assume itself as an interface articulating the public and private (exterior–interior), preserving a collective spirit.

Sagrado Coração de Jesus Church and the New Mosque of Lisbon present formal hypotheses and spatial solutions that instill this sense of interface. They represented objects that actively contribute to the composition of a space that is capable of valuing the architectural piece in itself but is also fundamentally capable of building mechanisms of relationships with the urban place in which they are situated. These objects start from an urban context, interpret it and draw a place (Vidler [1977] 1998). To this end, issues such as the introduction of a certain architectural porosity, where systems of connection between the exterior and interior are strongly promoted, diluting rigid limits and developing continuities between public and private space, are promoted (Benjamin 2019). On the other hand, the constitution of complex systems of approach and transition, sometimes even ambiguous, where one intuitively progresses between the public and private, collective and intimate, recognizing the various stages of a promenade characterized by moments of permanence, walking or passage, can be developed.

**Author Contributions:** Conceptualization, J.S.L., S.F. and C.D.C.; methodology, J.S.L., S.F. and C.D.C.; formal analysis, J.S.L.; investigation, J.S.L.; resources, J.S.L.; data curation, J.S.L.; writing—original draft preparation, J.S.L. and S.F.; writing—review and editing, J.S.L., S.F. and C.D.C.; visualization, J.S.L.; supervision, S.F. and C.D.C.; project administration, J.S.L., S.F. and C.D.C.; funding acquisition, J.S.L., S.F. and C.D.C. All authors have read and agreed to the published version of the manuscript.

**Funding:** This work is financed by national funds through FCT—Fundação para a Ciência e a Tecnologia, I.P., under the Strategic Project with the references UIDB/04008/2020 and UIDP/04008/2020, framework on the research project "Building Typology—Morphological Inventory of Portuguese City" with the reference code PTDC/ART-DAQ/30110/2017. In parallel part of the material it was also produce under the post-doc research with the with the reference SFRH/BPD/115838/2016 support also by FCT—Fundação para a Ciência e a Tecnologia, I.P.

**Institutional Review Board Statement:** Not applicable.

**Informed Consent Statement:** Not applicable.

**Data Availability Statement:** Data are contained within the article.

**Acknowledgments:** The authors acknowledge the architect Inês Lobo for providing detailed drawings of the New Mosque of Lisbon project. The graphics provided served as the basis for some of the analysis and interpretation presented in the article—Figures 8–10.

**Conflicts of Interest:** The authors declare no conflicts of interest.

## Notes

1. It is clear that the religious building serves as a support for the personal and individual reflection of human beings. Typologically, however, the religious building welcomes multiple people in this exercise of reflection. This fact, combined with the existence of collective religious ceremonies, promotes the meeting and constitution of an extended community based on the idea of the common good. Thus, the religious building is often a place of refuge for individuals, but it is also a place of sharing with a strong vocation for the collective use of space.

2. The research project Building Typology, produced by the research group *formaurbis* LAB at the Lisbon School of Architecture, is financially supported by the Foundation for Science and Technology, ref. PTDC/ART-DAQ/30110/2017, with the coordination of Remove for review. Further information: https://formaurbislab.fa.ulisboa.pt/the-building (accessed on 15 February 2024).

3. Nuno Teotónio Pereira (1922–2016), architect, is one of the most remarkable figures of his generation in Portugal and particularly in Lisbon, leading a modern architectural language with several brutalist references. He was one of the founders of the Movement for the Renewal of Religious Art, contributing several works which question the separation between public and private space through a rigid limit. Collective spaces find expression in several of his works, as a way of constituting transition systems between the building and the city. Nuno Portas (1934–...), architect, stands out as one of the most influential thinkers on the Portuguese city in the second half of the 20th century. He began collaborating with Nuno Teotónio Pereira in the 1950s and in 1958 began working with the magazine Arquitectura, where he would become director, and in which he wrote numerous texts that won him the Gulbenkian Prize for Art Criticism in 1963. After 1974, he played an important role in the production of social housing under the SAAL programme and, more recently, he stood out for his theoretical texts characterising extensive urbanisation and intervention in urban planning in the Ave valley region, in the northwest of mainland Portugal.

4. Inês Lobo (1966–...), an architect, represents a new generation of Portuguese architects who are recognised in Portugal and internationally. She has also worked as a curator and commissioner of architectural exhibitions, was responsible for the Portuguese representation at the 2012 Venice Biennale and was the Portuguese delegate to the VIII BIAU–Ibero-American Biennale of Architecture and Urbanism. She was a guest participant in the 2016 Venice Biennale, "Reporting from the front", and "Freespace" in 2018.

5. The post-doctoral research is being carried out by Remove for review at the Remove for review, with financial support from the Remove for review. The research focuses on the study of the intermediate space along a set of road mobility axes located in the Lisbon Metropolitan Area, understanding the processes of transformation of this interstitial membrane and simultaneously identifying the transformative value of the public space that certain buildings are able to play. In this sense, the research develops a parallel analysis of a set of reference examples, singular contemporary architectural objects, which constitute cases of particular interest in the transformation of the design of public space and the way we use the city. These reference cases are intended to constitute a conceptual basis for didactic support and reflection on the problem underlying the research as a whole.

6. The different studies that have been performed on building typologies focus mainly on classifying and ordering the case studies according to their uses, i.e., linked to the specificities of each functional programme. In the Building Typology Morphological Inventory of Portuguese City, it was decided that in addition to sorting the cases by their programme, there should be two other categories that sought to highlight the relationship between the building typology and the physical context (landscape

or urban), as well as building typologies that are the result of several transformation processes due to the action of time. The *Programme* category questions buildings in the articulation between the composition of the built forms and the function of space, produced from types that reveal the multiple possibilities of how architectural spaces that share the same purpose are organized in Portuguese territory. The *Context*, or the support understood as the inherent value to urban and architectural creation, condenses a set of physical, natural and human characteristics, which can be both inhibitors and drivers for the construction phenomenon, as well as interacting in the design process, acting directly about the programme in the composition of the building. This category includes buildings that show adjustment effects or in which the built form and the very organization of the architectural space are determined or profoundly influenced by the physiognomy of the territory, by the material that constitutes a particular site or even by the referential presence of natural elements. In the *Time* category, a reflection on the evolution of built forms is proposed, focused on a critical approach to the recycling potential of architecture. Here are grouped buildings that are the result of a process of successive modifications and whose shape is mainly influenced by the effects of the dynamic action of time. These buildings reveal the resistance of the built forms and are the product of the evolution to which they were subjected, bearing the scars of these mutations in their form. Usually, they are architectural objects that subsist adapted to different functions from those that originated them, and sometimes they are even buildings that are transformed to create other buildings.

[7] Each of the 120 buildings has been synthesized according to the same reading matrix. The cases are presented through plans, sections and elevations, an axonometric perspective, a territorial plan, photographs and a summary characterization text.

[8] It is important to note that more recent architectural examples seek to reinvent this more isolated sense of individual introspection, sometimes even without a direct and clear association with a particular creed, seeking to make the experience of reflection closer to the atmosphere itself than the architectural object creates, by itself, and/or establishes with nature. Take for example situations such as the Bruder Klaus Field Chapel (2007) and the Wooden Chapel (2018) designed by Peter Zumthor and John Pawson, respectively, or the Monte Chapel (2018) designed by Álvaro Siza.

[9] Shinto shrines have a system of pathways (Sando), often surrounded by natural spaces (forest or urban parks), which seek to build a progressive transition between the city and the temple. These paths form part of the overall temple enclosure but are for public use. They are places that extend the public space of cities or are interconnected with leisure routes and the enjoyment of the landscape. Although they are free to use, the paths seek to preserve the symbolic and mythical sense between mankind and nature, acting as a space of preparation and a moment of passage between the common and profane space and the sacred place. For more information on this topic, see Imazumi (2013). *Sacred Space in the Modern City. The Fractured Pasts of Meiji Shrine, 1912–1958*. Boston: Brill.

[10] "*… of spatial continuity and the tendency to erase every articulation between spaces, i.e., between outside and inside, between one space another. Instead I suggest articulation of transition by means of defined in-between places which induce simultaneous awareness of what is significant on either side. An in-between place in this sense provides the common ground where conflicting polarities can again become twin phenomena*". in Van Eyck, Aldo. 2008. *Writings. The Child, the City and the Artist*. p. 63.

[11] Religious Art Renewal Movement (*Movimento de Renovação da Arte Religiosa*).

[12] Nuno Portas testifies to this conceptual principle in the 2019 book "Nuno Portas: *18 obras partilhadas*" edited by Paulo Providência and Pedro Baía.

[13] "*Mouraria has a singular reality, were strong typicality and multi-culturality coexist, something that is a singular opportunity to achieve the effective integration of the OTHER, by sharing a common space: the square, a neighborhood or the public space in general*". In Lobo, Inês. "Mosque in Mouraria", in http://ilobo.pt/Mosque%20in%20Mouraria.html. Accessed on 6 January 2024.

[14] "*… with regard to buildings, it is a question of resuming the public vocation (...) and of creating new spaces associated with public programs at key points, which allow not only to create local dynamics but also to increase flows and experiences of this territory at the* scale of the city". In Lobo and Varela (2020). "Limite: identificação de um território". p. 16.

[15] Hagia Sophia was built to be the Cathedral of Constantinople, now Istanbul, in the 6th century, and became the reference building for the design of mosques in Turkey in the 15th century.

[16] Mimar Sinan (1490–1588) is considered the greatest architect of the Ottoman Empire and was responsible for the typological redefinition of the sacred space of Turkish mosques. His work was inspired and strongly influenced by the Hagia Sophia.

[17] Typologically, the courtyard in the mosque takes on a mostly private sense with more restricted access. The collective sense is present, but it is intended for the religious community.

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
