# Peer review of "The Sacred Building and the City: Decoding the Formal Interface between Public Space and Community"

_religions, doi:10.3390/rel15020246_

Round 1

Reviewer 1 Report

Comments and Suggestions for Authors

GENERAL COMMENTS

When you introduce an idea in one sentence, you need to explain it and provide more detail in the following sentences. Otherwise, the reader will not understand what you wanted to say and why you mentioned this idea in this particular place. You need to be clear.

When you mention an architect or building, you need to provide more information, not only names.

Figures: The font size is too small; explain the figure in the text with more detail.  

1.     INTRODUCTION

The Introduction lacks a theoretical framework with relevant references, as well as a clear statement of your aim and objectives. At the end of the Introduction, it would be good to explain the structure of the article so the reader would know what to expect.

Line 45: „collective sense“ – explain this

Line 63: it is not clear

2.     METHODS

You mention three theoretical categories but you do not explain any of them. Also, state why you chose to analyze these two cases (and not the other 118).

Most of the figures in the article are too small and insufficiently explained in the text.

Line 119: Explain what is morphological and spatial decomposition. Use the references and examples.

3.     SACRED – PUBLIC

In this subtitle, you mention hospitality, permanence and transition. Use the same words in the text and explain in more detail. For example, in line 150 you mention examples from ancient Greece or Japanese Shinto shrines but you need to elaborate on these examples. You can say that temples in ancient Greece, such as a temple of XX in XX, were built in XY way, forming ZY space that allowed XX relationship between sacred and public space.

Line 186: Interior-exterior / indoors-outdoors / inside-outside

5.     CASE STUDIES

Line 223: Give examples of echoes through various spatial relationships.

Figure 4: In the title of the figure is written Time-Space. What is Time? Explain in the text.

Line 230: What are the examples? Be clear.

Line 245: What is the way it connects to the city? Explain. Be clear.

6. DISCUSSION

Line 386: Hagia Sophia (built where, when, by who…why is it important to be used here as an example?)

Line 388: Mimar Sinan (when did he live, where, why is he important?)

Comments on the Quality of English Language

Proofreading is needed.

Author Response

Please see the article reviewed in attachment. 

Cover letter for review 01

GENERAL COMMENTS

. Corrections to the English and occasional adjustments for a better understanding of the narrative were made.

. biographical notes on the main architects mentioned and directly linked to the design of the buildings under study have been added to the endnotes.

. With regard to the font size of some of the internal image captions, we realise that they may be a little small, however, this is a digital publication and can therefore be zoomed in enough to be readable. On the other hand, we don't want the identification of the cases in the comparative tables to disturb the typological reading that the table produces. The cases in the tables are a sample and illustrate the diversity, formal and spatial richness related to the theme. The specific identification of each case assumes secondary importance. However, if you consider it vital to make these legends immediately legible, we can correct them for the final version before publication.

  1. INTRODUCTION

. some more theoretical references have been introduced

. "collective sense" - an endnote explaining the expression has been added .

. line 63 . corrected the meaning of the sentence

  1. METHODS

. The theoretical meaning of the categories is explained in an endnote 6

. Further bibliographical references supporting the concept of morphological decomposition have been added and the concept is explained in the following paragraphs.

  1. SACRED – PUBLIC

. The concepts host (formerly hospitality), permeance and transition were explained and duly included in the texts.

. The meaning of the Greek and Japanese examples has been clarified and supplemented with endnotes.

  1. CASE STUDIES

. Completed the legend of figure 4

. The identification of the cases present in figure 4 is in the image

. The explanation of how and why the mosque case is connected to the city is given a few paragraphs later in the sub-point dedicated to the characterisation of the mosque.

  1. DISCUSSION

. Additional information on the Hagia Sophia and Mimar Sinan placed at the bottom of the page

Reviewer 2 Report

Comments and Suggestions for Authors

1.      The paper would be strengthened in my opinion by disaggregating ‘community’ in order to explore the different communities that are involved in shaping urban spaces.

2.      This would require a more detailed outline of Lisbon’s changing social and cultural character and how the contested nature of urban space reflects those changes which are bound up with transnational migration and ethnic diversity, for example.

3.      The design of the Lisbon mosque raises the question of why it is being planned now, who are the Muslim supporters of the plan and who might oppose the plan. Given the opposition towards plans to establish churches and mosques in other European cities exploring this question would be very helpful to wider urban research. 

4.      The mosque design also raises the question of why the architects selected this particular design since it looks very plain and there does not appear to be any sign of a minaret or dome which in many cities around the world have come to symbolise mosques. The design looks similar to a proposal to build a mosque in London’s ‘East End’so it raises again the issue of comparison with similar proposals in other European cities. What does this paper tell us about religious buildings and urban space more generally then?  

5.      The abstract text is poor and needs improvement. The text in the main article is generally good but would be strengthened by using inclusive terms such as ‘people’ rather than ‘Man’ and ‘they’ rather than ‘he’.

Comments on the Quality of English Language

The abstract text is weak and needs improvement. The text in the main article is generally good but would be strengthened by using inclusive terms such as ‘people’ rather than ‘Man’ and ‘they’ rather than ‘he’.

Author Response

Please see the article reviewed in attachment. 

Cover letter for review 02

1. and 2.

. We recognise the value and interest of the observation made, however, we feel that this approach is not justified because it is a little dispersive for the subject of the article. The article focussed on issues of the form and urban / architectural space. How the shape of buildings can create spatial supports that will later guide the way people inhabit the city.

3.

Answered at the beginning of the case study characterisation.

4.

The comment made makes sense in part. In fact, the mosque has an architectural language that at first glance does not follow the classical and typological standards of a mosque. However, elements such as the minaret have been preserved, as can be seen in the section shown in figure 8, and spaces such as the courtyard (traditionally closed off) are re-invented in this project. The courtyard is transformed into a public square that opens up to a wider population, seeking to integrate the Muslim community with the rest of the neighbourhood. These particularities of the project are described in the text.

5.

The abstract was totally review.

Round 2

Reviewer 1 Report

Comments and Suggestions for Authors

No further comments.

Comments on the Quality of English Language

Check English for typographical errors.

Author Response

We have made some adjustments requested by the other reviewer, but nothing that questions the structure or objectives of the article.

Reviewer 2 Report

Comments and Suggestions for Authors

The paper makes a very interesting comparison between two types of religious buildings in a particular European city and has considerable potential to contribute to European urban studies, in general. Yet, since the authors are focusing on architectural form and urban space, there needs to be greater clarity about the difference between the ways they approach urban form and space and the ways in which social scientists in urban studies approach urban form and space, especially since urban sociologists, anthropologists and geographers have explored such issues as community, human meanings, emotional links, ritual etc in a variety of urban contexts. Readers can then locate the paper clearly within its specific disciplinary context. 

With regard to the specific issue raised about the design of the Lisbon mosque, Fig 8 does not made clear to the non-specialist that a minaret is envisaged. Fig 10 is more obvious to the non-specialist and there is no minaret in view. Indeed, the building looks very anonymous suggesting a particular choice about how to represent a mosque in a specific urban location. In London, for example, there have been very lively debates about the design of mosques, temples and churches which have been explored by social scientists. 

Comments on the Quality of English Language

The reservations in the Recommendations section could be answered to some extent by improving the English, The Abstract still needs considerable improvement while the English in the main text is still unclear in a number of places. Use needs to be made of someone whose English is their first language so that the text reaches the requisite standard for an international Anglophone journal. 

Author Response

In this second review, we once again try to clarify the questions posed by the reviewer. The English has been refined and clarified, and in the "discussion" section we have added a new explanation of the value that each of the buildings has in the construction of public space. How each building designs public space and how these characteristics are not bound by faith. With regard to the discreet presence of the minaret in the mosque project, we would like to reiterate that it seems clear to us that it exists in the section shown in figure 8 and that its absence in the illustration in figure 10 refers to a choice made by the project's author. The image was provided to us by the author herself and we cannot pass judgement on this graphic choice. Finally, with regard to the social issues raised, we emphasise that the aim of the article is to identify and highlight how certain formal characteristics contribute to the public or collective life of the urban space and how the building designs it. Even so, we reinforce the discourse by highlighting the existence of some urban uses in the mosque, such as the kitchen and the communal dining room, which go beyond the restricted use of the Muslim population, and which are open to the city's population in general